# NetR and AttR, Two New Bioinformatic Tools to Integrate Diverse Datasets into Cytoscape Network and Attribute Files

**DOI:** 10.3390/genes10060423

**Published:** 2019-06-01

**Authors:** Armen Halajyan, Natalie Weingart, Mirza Yeahia, Mariano Loza-Coll

**Affiliations:** 1Department of Biology, California State University, Northridge (CSUN), CA 91330, USA; armenhalajyan@gmail.com; 2Department of Computer Science, California State University, Northridge (CSUN), CA 91330, USA; natalie.weingart.567@my.csun.edu (N.W.); mirza.yeahia.904@my.csun.edu (M.Y.)

**Keywords:** genome-wide screens, genetic and protein interactions, biological networks, Cytoscape, *Drosophila* intestinal stem cells

## Abstract

High-throughput technologies have allowed researchers to obtain genome-wide data from a wide array of experimental model systems. Unfortunately, however, new data generation tends to significantly outpace data re-utilization, and most high throughput datasets are only rarely used in subsequent studies or to generate new hypotheses to be tested experimentally. The reasons behind such data underutilization include a widespread lack of programming expertise among experimentalist biologists to carry out the necessary file reformatting that is often necessary to integrate published data from disparate sources. We have developed two programs (NetR and AttR), which allow experimental biologists with little to no programming background to integrate publicly available datasets into files that can be later visualized with Cytoscape to display hypothetical networks that result from combining individual datasets, as well as a series of published attributes related to the genes or proteins in the network. NetR also allows users to import protein and genetic interaction data from InterMine, which can further enrich a network model based on curated information. We expect that NetR/AttR will allow experimental biologists to mine a largely unexploited wealth of data in their fields and facilitate their integration into hypothetical models to be tested experimentally.

## 1. Introduction

Recent technological advances have allowed researchers to produce a large amount of genomic and transcriptomic data in a wide range of experimental systems. This wealth of data is usually uploaded in their raw format to public repositories, such as the Gene Expression Omnibus (GEO) Database (https://www.ncbi.nlm.nih.gov/geo/), the European Bioinformatics Institute (EMBL-EBI) (https://www.ebi.ac.uk/), the DNA Data Bank of Japan (DDBJ) (https://www.ddbj.nig.ac.jp) and others, and can later be queried in a series of downstream applications and metanalyses by colleagues worldwide. While several tools exist for querying, retrieving and analyzing raw data deposited in public databases (e.g., a large collection of software packages distributed through the Bioconductor open source community; www.bioconductor.org) [1,2], the generation of system-wide data still vastly outpaces its re-utilization in subsequent studies. For instance, the number of GEO dataset deposits is nearly ten times higher than the number of publications using previously deposited GEO datasets (59,331 vs. 7160 according to NCBI, GEO Citations Listings: deposits and third-party usage, www.ncbi.nlm.nih.gov/geo/info/citations.html, accessed on 28 May 2019).

Such underutilization of data may result from a series of reasons. The scientific data management community has recently agreed that scientific data should be shared following four FAIR principles: Data should be Findable, Accessible, Interoperable and Reusable [3]. While the public repositories listed above may ensure relatively well the findability and accessibility of raw datasets, their interoperability often depends on different extents of dataset pre-processing to ensure that they share formatting and relevant entities map correctly between files [4]. In this regard, data reuse may be hindered by a pervasive lack of expertise in data management and processing by most experimental researchers [5]. While bioinformaticians have developed various outstanding methods for accessing and reusing public datasets, these methods are often difficult to implement for experimental biologists without a programming background. In fact, it has been argued that such requirement for specialized training in the use of dedicated bioinformatics tools may contribute to the gap between data production and analysis and, somewhat ironically, slow down the publication of findings [5].

Lastly, and perhaps most importantly, there exists an invaluable wealth of information in the form of processed data that is not easily ‘findable’ and/or ‘accessible’ and is therefore less amenable to automated analysis through bioinformatics (particularly by experimental life scientists with little or no programming experience). These include data that are often published as supplementary files and tables accompanying research publications, datasets that researchers may offer publicly through their own websites or upon request, data generated by online bioinformatics tools, or even extremely valuable data that may be unpublished but available to researchers in-house. Researchers with more advanced programming skills, or with easier access to collaborators who can provide bioinformatics support, can often generate simple *ad hoc* scripts that enable them to conduct a more exhaustive and careful integration of data from different sources and with heterogenous formatting. Unfortunately, however, many experimental biologists may be discouraged from attempting a more systematic integration of existing data given the time and effort involved in reformatting and integrating diverse datasets.

To facilitate the use of these valuable but less accessible data, we have developed two open source and easy-to-use computer programs (NetR and AttR) that allow users to combine diverse lists of genes or proteins into a network file that can later be mapped and visualized as a network of hypothetical interactions with Cytoscape [6].

Many high throughput technologies are used to identify lists of genes or proteins that interact with or are somewhat targeted by a core gene (or protein) of interest. Examples may include genes that are differentially expressed following the experimental manipulation of the core gene (e.g., by overexpression, knock-down, mutation, etc.), the identification of putative transcriptional targets based on DNA-binding profile of a core gene (e.g., by chromatin immunoprecipitation and sequencing (ChIP-seq) or binding site search within promoter/enhancer regions), the identification of protein interactors of a protein of interest by a large-scale proteomic screen (e.g., immunoprecipitation followed by mass spectrometry analysis, yeast two-hybrid screens, etc.), the identification of genetic interactors by a genetic screen, or a screen for putative targets of an enzyme (e.g., by phosphoproteome analysis, peptide sequence search, etc.). Any of these unidimensional lists of targets or interactors for a core gene or protein can be conceptualized as a wheel network, where the core gene is at the center of the wheel and the rim of the wheel represents its corresponding targets or interactors (Figure 1a). For example, a ChIP-seq experiment for a core transcription factor will generate a list of its putative target genes based on their proximity to the identified DNA-binding peaks. The obtained list of putative targets can be represented as a wheel network with the transcription factor at its hub and all the putative targets at the rim of the wheel. Likewise, knocking down the expression of a given cytoplasmic kinase by RNA interference, followed by transcriptome profiling through RNA-sequencing (RNA-seq) will generate a wheel network with the targeted kinase at its hub and all the differentially expressed genes at the rim. When multiple lists are combined, any targets that are shared among different wheel networks will become nodes that connect them into a larger network (Figure 1b and Appendix A). Such an approach can allow for a simpler and quicker visual identification of a smaller set of genes with hypothetically key regulatory roles, and NetR allows users to automatically combine any number of unidimensional lists into integrated network files that map all of the interactions between nodes in the combined network. The output produced by NetR is a comma-separated values (CSV) table which is easily readable by the open-source network visualization program Cytoscape [6].

In addition to visualizing the connections between various nodes in a network, Cytoscape also allows users to import node attributes, such as membership to a gene/protein family, expression levels, cellular localization, etc. Once imported, these attributes can be used to identify nodes with specific attributes based on different colors, shapes, sizes, etc. AttR allows users to generate an all-in-one attribute table containing all the attributes for nodes in a combined NetR network, with up-to-date gene identifiers and properly formatted to be used with Cytoscape. Here we share and demonstrate how to use NetR and AttR, and how their use enabled us to integrate valuable data that were publicly available but inaccessible to other bioinformatics tools. We also show how their use allowed us to identify a subset of genes statistically enriched for biological features that make them interesting candidates for further studies, and that would have likely remained unidentified using classical approaches to data metanalysis.

## 2. Materials and Methods 

### 2.1. Installing NetR, AttR, and Their Dependencies

A step-by-step description of how to install NetR and AttR, as well as the necessary dependencies to run both programs is provided in the supplementary methods file (Appendix A; *Supplementary Methods*). NetR and AttR were coded in Python3, and both scripts can be executed from a command terminal. The NetR script (Appendix A) and AttR script (Appendix A) are available as supplementary material, and have also been deposited in the open-source software sharing repository GitHub (https://github.com/armenhalajyan/NetRAttR). and in the GitHub ReadMe document. Once both programs and dependencies have been downloaded locally, users can navigate to a directory containing both files and run either program by entering “python3 NetR.py|AttR.py” (Mac OS) or “py NetR.py|AttR.py” (Windows), respectively (Appendix A). Starting the programs will open the corresponding program graphical user interface (GUI) (Figure 2a and Figure 3a), both of which function through a familiar look and operability. Alternatively, Windows users can download executable files for both programs from the same GitHub repository. Double-clicking on the executable files (or the corresponding program icons if the files were moved to the desktop) will start the programs as usual (NB: When using the executable files for the first time, users will get a standard message about opening a program from an unverifiable/untrusted source).

### 2.2. Processing Files for Use with NetR

The sample datasets used to demonstrate the use of NetR and AttR are available as supplementary files, both in their original versions as obtained from their corresponding source, as well as processed CSV files ready to be used with NetR and AttR.

The list of putative targets of Esg in intestinal cells as identified by DamID was originally published as an MS Excel file in the original article [7]. The original file listed 2327 putative targets of Esg, as identified in one or more of the DamID technical replicates (Appendix A; *Esg_DamID data-original.xlsx*). This original file was processed to contain only putative targets identified in all three technical replicates (see Appendix A; *Supplementary Methods* for more details). The processed data were then saved as a CSV file and contained only 1071 genes from the original target list (Appendix A; *Esg_DamID data-processed.csv*). We then processed these data further, by calculating the average signal-to-background ratio across technical replicates and ranking the averages to obtain a list of the top 10 Esg targets used to illustrate the integration of InterMine database (http://intermine.org/) data (Appendix A; *Esg_DamID data-top10.csv*).

We also used supplementary data from a report by Jin and collaborators, who used DamID in whole midguts to generate a list of putative Capicua binding sites [8] (Appendix A; *Cic_DamID data-original.xlsx*). The original MS Excel file was processed as described in Supplementary Methods to retain only the data related to Cic genome mapping and saved as a CSV file (Appendix A; *Cic_DamID data-processed.csv*). This file was then further processed to obtain a list of the top 10 Cic targets used to illustrate the integration of InterMine data (Appendix A; *Cic_DamID data-top10.csv*).

Lastly, the list of genes differentially expressed in intestinal stem cells following the overexpression or knockdown of the RNA-binding protein Tis11 was obtained from two supplementary MS Word files associated with the original article [9] (Appendix A; *Tis11-downregulated-original.docx* and *Tis11-upregulated-original.docx*). These data were processed as described in supplementary methods and saved as a CSV file (Appendix A; *Tis11-diff expressed-processed.csv*).

#### 2.2.1. Uploading Datasets to NetR (Esg-Cic-Tis11 Network)

The Appendix A
*Esg_DamID data-processed.csv* was first uploaded to NetR, with “esg-DamID” as the Dataset name, “*Drosophila melanogaster*” as Organism, “esg” as Core gene symbol and “DamID” as Technique (and as a file with column headers). On the file preview window, the Fly Base ID column was chosen for NetR processing (bolded). Clicking Okay prompted NetR to ask if InterMine interaction data should be integrated into the NetR network, which was denied. Next, the Appendix A
*Cic_DamID data-processed.csv* was uploaded as a file with headers, with “cic-DamID” as the Dataset name, “cic” as Core gene symbol, “DamID” as Technique and choosing the GeneID column in the preview window. Lastly, the Appendix A
*Tis11_Diff expressed-processed.csv* was uploaded as a file with headers, with “Tis11-DEG” as the Dataset name, “tis11” as Core gene symbol, “RNAseq” as Technique and choosing the FlyBaseID column in the preview window. The obtained NetR network file was saved as *EsgCicTis11_noIM.csv* (Appendix A). More detailed, step-by-step instructions on the use of the NetR GUI can be found in supplementary methods.

#### 2.2.2. Uploading Datasets to NetR (Esg-Cic Network + Intermine Interactions)

The Appendix A (*Esg_DamID data-top10.csv*) was uploaded as a file with column headers to NetR, with “esg-DamID” as the Dataset name, “*Drosophila melanogaster*” as Organism, “esg” as the Core gene symbol and “DamID” as Technique. On the file preview window, the Fly Base ID column was chosen for NetR processing (bolded). The integration of InterMine interaction data was accepted this time. Next, the Appendix A (*Cic_DamID data-top10.csv*) was uploaded as a file with headers, with “cic-DamID” as the Dataset name, “cic” as Core gene symbol, “DamID” as Technique and choosing the GeneID column in the preview window. The obtained NetR network file was saved as *Esg_Cic-top10_IM.csv* (Appendix A).

### 2.3. Processing Files for Use with AttR

As a first set of attributes for the NetR networks, we used data from Dutta et al. [10], who profiled gene expression by RNA-seq from different cell types and from different regions of the *Drosophila* intestinal epithelium, under homeostasis or following infection (GEO Accession # GSE61361; Appendix A; *ISC_RNAseq-original.txt*). The original dataset was downloaded in .TXT format, opened directly in MS Excel and processed as explained in supplementary methods. Briefly, a new column was inserted and used to calculate the average across physiological ISC RPKM values. The first four columns (GeneID, ANNOTATION_SYMBOL, NAME, and the new column for ISC averages) were then copied, pasted as values into a new workbook (Appendix A) and saved as a CSV file (Appendix A; *ISC_RNAseq-processed.csv*).

In addition to the Discrete/Continuous attributes table from RNA-seq data described above, a List attributes file was generated based on a classification of *Drosophila* genes under the Biological Process “Intestinal stem cell homeostasis” for *Drosophila melanogaster* in the Gene Ontology Consortium website (http://www.geneontology.org/) [11,12], as described in supplementary methods. A list of 30 genes was obtained and saved as a TXT file from the browser, imported into a new MS Excel workbook and saved as a CSV file (Appendix A; *GO_IntestinalHomeostasis.csv*).

### 2.4. Uploading Datasets to AttR 

The AttR program was started from a command prompt terminal as described in Appendix A. In the AttR GUI (Figure 3a), *Drosophila melanogaster* was selected as Organism, and the supplementary files *EsgCicTis11_noIM.csv* (Appendix A) and *ISC_RNAseq-processed.csv* (Appendix A) were selected as NetR Table and Attribute Table, respectively. The attributes file was selected as a Discrete/Continuous values table with column headers. In the file preview window shown by AttR after clicking Submit, GeneID was selected as the mapping column, and the header “Name” was deleted from the third column’s text field, excluding the column from further AttR processing. Next, the Appendix A (*GO_IntestinalHomeostasis.csv*) was uploaded as a List attributes file without headers and “ISC Homeostasis” was added to the first column text entry field (Figure 3c), making it the only column that was further processed by AttR. The CSV file created by AttR was named *ISCattributes-EsgCicTis11net.csv* and is available as a Appendix A.

To generate an AttR attributes file for the Esg-Cic network that integrates InterMine interaction data, the same steps described above were followed, with the exception that *Esg_Cic-top10_IM.csv* (Appendix A) was selected as the NetR Table file. The AttR file obtained was saved as a CSV file (Appendix A; *ISCattributes-EsgCicIMnet.csv*)

### 2.5. Mapping NetR Networks in Cytoscape

The *EsgCicTis11_noIM.csv* (Appendix A) was opened in Cytoscape v3.7.1. The “Source Symbol” was designed as the “Source Node” column (green dot icon); “Interaction Type” was selected as the “Interaction” column (blue arrowhead icon); and “Target Symbol” was selected as the “Target Node” column (red target icon). The “Source Primary Identifier” and “Source Secondary Identifier” were assigned the “Source Node Attribute” column types (green text icon), whereas “Target Secondary Identifier” and “Target Primary Identifier” were assigned as “Target Node Attribute” columns (red text icon), respectively (Appendix A). Clicking “OK” opened the network. By default, the network did not show node labels; they were displayed via View > Show Graphic Details (Appendix A). The network was then manually re-arranged as explained in supplementary methods, to generate a map that highlights the clusters of putatively mono-, di- and tri-regulated targets (Appendix A).

Next, the *ISCattributes-EsgCicTis11net.csv*
Appendix A generated by AttR was imported as a node attributes file (File > Import > Table from file), with the appropriate “Mapping Key” automatically pre-selected. To apply the attributes to nodes and interactions, the node and edge styles were edited in the Style tab as follows: For nodes, Fill Color:Mapping Type was set to Continuous Mapping, and the newly imported “ISC” column was selected for mapping (Column:ISC). The limits of the fill color range were set to 0–50 for the attribute (RPKM values from RNA-seq data), and the colors were set to “99FF99” (light green) and “006600” (dark green), respectively by double-clicking the corresponding arrow. Genes that were not present in the dataset from Dutta et al. [10] were forced to display the light green color for no/low expression by setting the Default color code to “99FF99” (Appendix A). To allow for a better contrast of the node labels, the settings for Label Color were also modified as follows: Mapping Type and Column were set to “Continuous Mapping” and “ISC” respectively, as before. In the Continuous Mapping Editor for Node Label Color, the Set Min and Max range was again set to 0–50. Double-clicking on the leftward arrowhead for “below color” and on the downward arrowheads at the 0 and 25 positions, the label color was set to black (Colors > RGB > Color Code “000000”). Similarly, the downward arrowhead at position 50 and the rightward arrowhead for “above color” were set to Color Code “FFFF00” (yellow). Lastly, the Shape parameters were modified in the Control Panel to incorporate the GO classification under Intestinal stem cell homeostasis: Control Panel > Style (Nodes) > Shape > Mapping Type: Discrete; Column: ISC Homeostasis; false: Round Rectangle; true: Ellipse.

To obtain the network image shown in Figure 4a, two final display modifications were applied: (1) Each cluster was selected manually and changed to a grid layout (Layout > Grid Layout > Selected Nodes Only—Appendix A); (2) the *esg*, *cic* and *Tis11* nodes were enlarged for illustration purposes. Each node was selected using the Search window, and their default display parameters for Height, Width and Label Font Size were bypassed under the Style tab (Height = 140; Width = 450 and Label Font Size = 240).

### 2.6. Filtering Out Terminal Nodes and Linear Paths

The *Esg_Cic-top10_IM.csv* NetR file (Appendix A) was imported to Cytoscape (Appendix A). Duplicated edges in the network were removed and a 2-step node filtration was performed to discard any terminal or intermediate nodes (see Appendix A for more details). Briefly, nodes with in/out degrees equal to or greater than two were selected and used to create a new network, which effectively removed all terminal nodes from the original network (Appendix A). The same filtration step was then repeated on the intermediate network to filter out nodes that originally linked a core gene and terminal nodes and had themselves become terminal nodes in the intermediate network (Appendix A). The *ISCattributes-EsgCicIMnet.csv* AttR file (Appendix A) was then imported and the node display and label color modifications described above were used to color nodes based on their ISC expression and GO classification (Appendix A). To obtain the network in Figure 5b, nodes were manually rearranged, and the Edge Style parameters were modified as follows: Control Panel > Style > Edge > Stroke color > Mapping Type: Discrete Mapping; Column: Interaction; DamID: RGB 0000FF; genetic: RGB 00FF00; physical: RGB FF00FF.

### 2.7. Statistical Analysis

Hypergeometric tests were conducted using the dhyper function in an *ad hoc* R script provided here as a Appendix A (*50+esg-full-cic-tis11 no intermine.R*). The following parameters were used: Number of *Drosophila* genes, 15682; number of genes connected to each core gene, 1072 (*esg*), 3692 (*cic*) and 309 (*Tis11*); number of genes in the whole network, 4459; and number of putative co-regulated targets, 23. The number of “highly expressed” (RPKM >= 50) [10] or ”not highly expressed” genes in each set were as follows: *Drosophila* genome, 1251/14431; whole Esg-Cic-Tis11 network, 608/3850; *esg* set, 156/1415; *cic* set, 550/3142; *Tis11* set, 37/271; and putative co-regulated targets, 8/15. A detailed explanation of how these values were obtained is provided in supplementary methods (Appendix A).

## 3. Results

### 3.1. Using NetR to Integrate Datasets (No InterMine Mode)

In order to illustrate the operation of NetR, we use three sample datasets from open access articles related to the genetic regulation of intestinal stem cells in the model organism *Drosophila melanogaster* (all of which are available as supplementary files—see methods section and Appendix A for more details). First, we use an MS Excel file that contained the full list of putative transcriptional targets of the *D. melanogaster* transcription factor Escargot (Esg) in intestinal cells, based on their proximity to sites where Esg bound intestinal DNA, as revealed by DamID [7] (Appendix A). Second, we use an MS Excel file that listed the putative transcriptional targets of Capicua (Cic), as identified in a similar manner by DamID [8] (Appendix A; *Cic_DamID data-original.xlsx*). Lastly, we use two MS Word files that listed intestinal stem cell mRNAs bound by the RNA-binding protein Tis11 [9] (Appendix A). We intently chose these datasets as examples to illustrate the operation of NetR and AttR because they nicely represent the type of data for which NetR and AttR were originally conceived (i.e., processed data that are publicly available, yet not properly formatted to be readily usable through existing network mapping programs).

NetR uses files that organize data in one or more columns, with simple or no headers. Occasionally, data tables obtained from publications, websites or internal laboratory resources contain additional information in them or may have slightly different formatting. Preparing files to be used with NetR and AttR, however, involves only a few copying and pasting steps. In the examples provided here, the original Appendix A required some minor pre-processing to generate properly formatted CSV files (Appendix A—see methods section and Appendix A for details).

When users start NetR, they will see a GUI like the one shown in Figure 2a, where they will have to name a dataset to be uploaded, (e.g., “Esg-DamID”), choose the CSV file to be uploaded (e.g., Appendix A; *Esg_DamID data-processed.csv*), choose an organism name (e.g., “*Drosophila melanogaster*”), identify the name of the core gene or protein (e.g., “esg” and “DamID”, respectively). It is important to type in a correct gene symbol for the core gene or protein. NetR will use the typed symbol to communicate with the InterMine database, and if the symbol is not properly spelled the program cannot proceed, forcing a re-initiation of the process.

A checkbox labeled “Check if the file has a header” allows the user to determine if the values in the first row of a list or table should be used as column headers and excluded from the data. The last row contains standard “Reset”, “Clear”, and “Submit” buttons, which allow users to restart the program and discard any submitted datasets to that point, clearing the entries for the current dataset, or upload the dataset, respectively.

Clicking on the Submit button opens a preview of the selected CSV file, displaying column headers and the first five rows of the table (displayed as read-only; Figure 2b), where users can select the column(s) containing the relevant data. When the desired column(s) is(are) selected, the user clicks “Okay” and NetR uploads the chosen data into the program. As part of the first data upload, a dialog box will ask users if they want to integrate interaction data from InterMine (Figure 2c), which users will have to answer only once. Integration of InterMine data is explained in further detail in a separate section below.

Once a dataset has been uploaded into NetR, a standard dialog box will ask users if they would like to upload additional datasets. If the user answers “Yes”, NetR will clear all fields and allow the upload of a new dataset, displaying the datasets uploaded thus far (e.g the NetR GUI will appear as in Figure 2d after uploading the data for Esg and Cic, and just prior to uploading the data for Tis11). When a user has uploaded all desired datasets and answers “No” to uploading additional datasets, NetR will generate a standard “Save as” window prompting the user to name and route the CSV file generated by the program. In our example, each of the CSV datasets were submitted to NetR without integration of InterMine interaction data, and the resulting NetR output file is provided as Appendix A
*EsgCicTis11_noIM.csv*.

### 3.2. Downloading, Installing, and Using AttR

In addition to mapping connections between nodes in a network, Cytoscape allows users to display nodes and connections in a network in ways that reflect their specific attributes. Therefore, we developed AttR, a program that creates attribute files related to all the nodes present in a NetR file. The combined use of NetR and AttR allows users to create networks of interacting genes and proteins from a series of individual datasets, as well as display attributes obtained from public repositories, published articles, or their own unpublished data.

Datasets may include a wide range of attributes and can be broadly classified into two different types: List datasets and Discrete/Continuous datasets. List datasets are lists of gene/protein identifiers that share an attribute. For example, a list of genes that code for transcription factors, list of genes differentially expressed in a particular cell type, or even a list of genes for which specific reagents exist (antibodies, mutant lines, pharmacological inhibitors, etc.). Discrete/Continuous values refer to datasets in which genes/proteins are associated with one or more discrete or continuous values. Examples of a discrete values dataset could be “cellular localization” (e.g., “nucleus”, “cytoplasm”, “cell membrane”, “mitochondria”, etc.) or “molecular function” (e.g., “kinase”, “transcription factor”, “channel”, “phosphatase”, etc.). An example of a continuous attribute could be related to “expression level” or “expression change” in a given sample (i.e., RPKM values from an RNA sequencing experiment, or fold change in microarrays). For our example, we first used data obtained by RNA sequencing from isolated intestinal progenitors by Dutta et al. [10]. The original data were available through the NCBI GEO database (Accession # GSE61361; Appendix A, *ISC_RNAseq-original.txt*) and were processed as described in methods section and Appendix A to generate a properly formatted CSV attributes file to be used with AttR (Appendix A, *ISC_RNAseq-processed.csv*). We also obtained a list of *D. melanogaster* genes classified under the Biological Process “Intestinal stem cell homeostasis” by the Gene Ontology Consortium [11,12], through a series of simple steps described in methods (Appendix A; *ISC_RNAseq-processed.csv*).

The AttR GUI is shown in Figure 3a. Users need to choose an “Organism” from a drop-down menu (as in NetR); select the NetR CSV file to which they want to assign attributes; select the corresponding attributes CSV file; determine whether the attributes dataset to be uploaded is in List or Discrete/Continuous Values format, respectively; and indicate if the attributes dataset to be uploaded has column headers or not. The bottom row of the AttR GUI contains “Reset”, “Clear” and “Submit” buttons that function similarly to those in NetR.

When users click Submit, AttR opens a preview of the selected attribute CSV file, displaying the first few rows of the uploaded table as read-only. The file preview windows will look slightly differently, depending on the type of attributes dataset selected by the user (Figure 3b,c). In either case, users need to determine which columns with genes/proteins (for List attributes) or which columns containing attribute values (for Discrete/Continuous attributes) they wish to upload to AttR (see methods section and Appendix A for more details).

When the user presses “Okay”, AttR processes the uploaded data and subsequently asks users whether they would like to add another attribute table. Answering “Yes” will reinitiate the upload process for each additional attribute file, while answering “No” will allow AttR to combine all the uploaded attributes datasets, updating the submitted gene/protein identifiers through InterMine, and generating an attributes table in CSV format. A file dialog window will ask the user to name and choose a location to save the output CSV file. In our example, we generated the AttR file called ISCattributes-EsgCicTis11net.csv.

### 3.3. Example of NetR/AttR Outcome (without InterMine Data)

NetR and AttR generate files to be used with Cytoscape, an open source software for the mapping and visualization of networks of nodes and interactions with diverse attributes [6]. We opened the *EsgCicTis11_noIM.csv* NetR output file in Cytoscape, imported the *ISCattributes-EsgCicTis11net.csv* (Appendix A) as an attributes table and made the adjustments described in Methods to generate the network shown in Figure 4a.

By visually inspecting this network, we noticed what appeared to be a proportional enrichment of dark green nodes (i.e., highly expressed in ISCs according to data from Dutta et al. [10]) among the nodes targeted by all core genes, compared to other subsets of nodes in the network (Figure 4b). To formally test this idea, we created a column filter in Cytoscape that allowed us to obtain the numbers of genes with RPKM values equal to or greater than 50 among different subsets of nodes (all nodes, or nodes targeted by Esg, Cic or Tis11, or only the nodes that were targeted by all 3 core genes). We then used a hypergeometric test to compare the relative fractions of RPKM > 50 genes within the original dataset from Dutta et al. [10] to those in each of the relevant network subsets. We found that there was a significant enrichment of genes highly expressed in ISCs in each of the core gene wheels (Cic-wheel, *p* = 1.78 × 10^−62^; Esg-wheel, *p* = 1.86 × 10^−14^; Tis11-wheel, *p* = 0.0033), as well as a further enrichment of highly expressed genes in the subset of the 3-way shared targets compared to the wheel networks combined (*p* = 0.0064—see methods section for details).

### 3.4. Integration of Intermine Data

When NetR integrates two or more wheel-networks, they will appear connected by target nodes that are shared between two or more of the core genes or proteins in each of the integrated wheel-networks (e.g., genes targeted by two or all three of the intestinal master regulators from our sample data; Figure 4a). However, given the nature of the data used to build each wheel network, these modules will often contain only inbound regulatory connections for the shared targets (Appendix A). For instance, in the network shown in Figure 4b, all of the genes in the 3-way co-regulated set are putatively regulated by Esg, Cic and Tis11; however, we have not learned anything about how any of those shared targets could feedback onto (and potentially regulate) any of the core genes or any other genes in the network. In order to exploit public data that may reveal such putative cross-regulatory interactions between nodes in a combined network, NetR allows users to import additional genetic and protein interaction data available through InterMine [13,14]. While uploading the first dataset of a NetR session, users are asked if they would like to integrate interaction data from InterMine (Figure 2c). When they answer “Yes”, NetR will automatically query InterMine for any known interactions for all of the genes in the current and subsequent datasets (i.e., all of the nodes in a combined NetR network). Integrating these additional InterMine data will create outbound connections for several of nodes in a combined network (Appendix A), which in many cases will target other nodes within the same network. This approach allows for the identification of more complex regulatory modules of interacting genes within the network, with a mix of inbound and outbound links among nodes and potential regulatory loops (Appendix A).

To illustrate the integration of InterMine data into a NetR network, we used lists that include only the top 10 putative targets of Esg and Cic (Appendix A; *Esg_DamID data-top10.csv* and *Cic_DamID data-top10.csv*). Both reduced target lists were submitted to NetR, accepting the integration of InterMine interaction data (Figure 5a), which generated a new NetR file that incorporates known genetic and physical interactors for Escargot, Capicua and their combined top 10 targets (Appendix A; *Esg_Cic-top10_IM.csv*). This new NetR output file was then uploaded to AttR, along with *ISC_RNAseq-processed.csv* (Appendix A) and *GO_IntestinalHomeostasis.csv* (Appendix A), to generate a new AttR attributes file applicable to the new NetR network (Appendix A; *ISCattributes-EsgCicIMnet.csv*).

As expected, integrating InterMine interaction data to a NetR file greatly increases the number of nodes and connections in a network (Appendix A). However, when integrating InterMine connection data, we were mostly interested in potential feedback loops within the network. We therefore used Cytoscape functions to filter out from the network any “terminal” nodes and linear paths (i.e., those nodes and short paths that do not loop back to other nodes in the network), which generated a reduced network of only 25 highly interconnected nodes shown in Figure 5b (see methods section for details). A Gene Ontology enrichment analysis of the 25 genes in this new network showed a significant enrichment for genes classified under the Biological Process “Intestinal stem cell homeostasis” (*p* = 1.45 × 10^−5^; Fisher’s exact test with Bonferroni correction).

## 4. Discussion

Recent technological advances have allowed researchers to routinely produce large amounts of genome-scale data from a wide range of experimental systems. While several tools exist for querying, retrieving, analyzing and sometimes integrating public datasets, the generation of system-wide data generally outpaces its re-utilization in subsequent studies, most notably among the same experimental biologists who generated the data in the first place. Such data underutilization may in part result from the lack of consistency in the way that datasets are tagged, stored and annotated [15]. The latter may be particularly true about processed datasets that are available through publication, which are often hard to find through literature searches and are not easily analyzable through standard bioinformatics approaches due to data formatting inconsistencies. Furthermore, even when researchers are aware of several genome-scale studies by colleagues in their field, they may still be discouraged from more carefully re-analyzing them if they lack the programming skills that are often required to reformat and process public datasets.

Here we introduce two free and easy-to-use computer programs (NetR and AttR) that allow users to combine diverse lists of genes or proteins into networks that can be mapped with Cytoscape [6]. While a certain degree of data processing was still required to use NetR and AttR, they involved simple downloading, copy/pasting and sorting operations with familiar computer programs and internet browsers, all of which are much better aligned with the computer skills of most experimental biologists.

To demonstrate the features and applicability of NetR and AttR, we combined three independent datasets related to the genetic regulation of intestinal stem cells in the model organism *Drosophila melanogaster* [7,8,9]. Our approach allowed us to identify a small subset of targets that are putatively co-regulated by all three of the chosen master regulators in these cells (Figure 4b), which makes them interesting candidates for validation and further experimental work. Interestingly, there was almost no overlap between the identified set of putatively co-regulated targets and a similar set of candidates that would have been chosen from each original dataset based on more traditional data mining approaches (e.g., significance ranking). When each original dataset was ranked based on their corresponding signal intensity (peak intensity ratio for Esg and Cic targets, and a ratio of fold-change in expression for Tis11), and the 23 top targets from each set were compared to the 23 putatively co-regulated targets identified by NetR, there was only one target (bun) shared between the Cic and NetR sets. Therefore, integrating different datasets allowed us to identify a vastly novel set of interesting candidates for further studies. Furthermore, the subset identified with NetR was significantly enriched for genes highly expressed in intestinal cells, as determined by an unrelated study [10], as well as genes curated under the “intestinal stem cell homeostasis” by the Gene Ontology Consortium. Of course, out data await both validation (i.e., to independently confirm that the putatively co-regulated targets are indeed regulated by each of the master regulators) and further functional studies (i.e., to determine whether they play a functional role in regulating intestinal stem cells in the fly intestine). However, if we consider expression as a proxy for the likelihood that a candidate might play a regulatory role in a cell of interest, then our findings would indicate that our intersection of independent but related datasets enhanced the discovery rate of biologically significant targets relative to the individual studies that contributed to the intersection. Similarly, integrating InterMine data when creating a NetR network allowed us to identify a set of 25 highly interconnected genes (Figure 5b), which were also significantly enriched for genes classified under “Intestinal stem cell homeostasis” by Gene Ontology. Interestingly, 15 of these 25 genes were not even present in the original datasets, which represents an even clearer example of exploiting existing public data for identifying priority candidates for further studies. Our observations above, while interesting and encouraging, should be considered as anecdotal. We have not repeated our analyses with enough additional datasets to more confidently determine how often NetR/AttR lead to the identification of pursuable candidates that are both novel (i.e., that were not identified in the original datasets via traditional methods) and enriched for a biologically relevant feature. In this regard, it is important to emphasize that, *sensu stricto*, neither NetR nor AttR represent discovery tools (i.e., neither program generates new knowledge based on available data). Instead, we prefer to think of NetR and AttR as purely heuristic tools, i.e., tools that facilitate the formulation of testable hypotheses based on existing data. For instance, when we state that NetR allowed for the identification of putatively co-regulated targets of Esg, Cic and Tis11, we are inherently reckoning that they still require experimental validation.

Of course, other tools exist for the mapping, visualizing and analyzing genetic or protein networks, many of which also offer users the opportunity to integrate interaction data from a series of existing sources and biological species. For instance, the Agilent application within Cytoscape allows users to submit a list of molecular entities that are used by multiple scientific literature search engines to retrieve documents in which the query terms are associated to other entities through specific association terms of interest. The associations detected by the Agilent app are then collected into a Cytoscape network, in which the sentences used to support each association are stored as edge attributes [16]. OmicsNet (https://www.omicsnet.ca/) allows users to upload lists of molecular identifiers and map retrieved molecular interactions of 4 different types (protein-protein, transcription factor-gene, miRNA-target gene and metabolite-protein) [17]. In EsyN (http://www.esyn.org/), users can upload a list of genes or proteins from diverse species and generate network models based on genetic and protein interaction data retrieved from InterMine and BioGrid (https://thebiogrid.org/)—i.e., using a similar approach to integrating InterMine data when uploading gene lists to NetR [18]. Lastly, GeneMania (https://genemania.org/) (which can also be installed as an optional App in Cytoscape) offers a very powerful and user-friendly interface to find genes that are related to a list of genes provided by the user, based on association data that include protein and genetic interactions, curated pathways, co-expression and co-localization data [19]. Each of these, and additional resources, represent different approaches to mining publicly available data based on diverse association criteria. We propose that NetR/AttR complement these powerful tools in three main ways.
They offer an opportunity to integrate publicly available or unpublished data that exist in formats that are less amenable to database cross-referencing or querying by literature search engines.In the process of preparing datasets for uploading into NetR/AttR, users can exercise some degree of control over what nodes will be integrated into the network (e.g., by deciding on cutoffs of significance or selection criteria)Users can use their expertise and judgement to integrate datasets that cross-referencing algorithms may miss, based on dataset annotation.

In this regard, NetR/AttR should be considered as complementary to other biological data management and processing tools, providing a somewhat higher degree of control over the data being integrated into a network model (although it should be emphasized that some of the resources mentioned above offer post-mapping network editing features that allow users to discard nodes or edges based on filtering criteria).

Each of the tools mentioned above presents strengths and limitations. Perhaps more interestingly, they all make use of different datasets, repositories and retrieval protocols, which gives rise to non-overlapping network models. For instance, when *esg*, *cic* and *Tis11* were used as lists of query genes for Agilent, EsyN and GeneMania, the network models retrieved were not highly similar to each other, nor to the one that we generated using NetR/AttR. Most notably, the Agilent/Cytoscape, EsyN and GeneMania networks did not include the experimental data generated by our source studies [7,8,9], which are publicly available but not accessible to their cross-referencing algorithms. Thus, we believe that NetR and AttR can fill a significant gap in our ability to generate new testable hypotheses based on published observations, which could be deemed as important to scientific progress as the generation of new data per se.

One perceived limitation of NetR and AttR is that both are separate programs that simply produce network and attribute files to be used with Cytoscape. It is therefore reasonable to imagine that both programs would be more useful if directly integrated into Cytoscape as applications. However, we believe that Cytoscape integration would limit the flexibility and applicability of the programs, both of which were conceived and created to be open-source and easily customizable to serve additional purposes. While the NetR and AttR outputs work well with Cytoscape by design, the data in them could be useful outside of Cytoscape, particularly for more advanced users that may want to integrate them into their own bioinformatics pipelines. Furthermore, more advanced users may want to consider introducing the REST API into the NetR and AttR code, in order to call on powerful Cytoscape functions to work on the NetR and AttR files as part of their independent bioinformatic pipelines. In summary, we hope that more advanced users in the research and bioinformatics community feel compelled to use, experiment with, and improve NetR and AttR, in any number of unexpected and creative ways, including but not limited to full Cytoscape integration.

Ten years ago, while reflecting on a minimal bioinformatics skill set that modern biologists should possess, Tan et al. wondered how undergraduate and graduate life science students, already charged with having to learn the fundamentals and latest developments in various subdisciplines, could find the time to cope with additional bioinformatic and programming demands [20]. These challenges are even more true nowadays, especially when the ease and speed with which new data are generated make it harder for students to keep up with the ever-expanding developments in modern biology. The constant advances in “-omics” techniques also make it increasingly challenging, even for trained bioinformaticians, to keep up with the diverse data structures and mathematical frameworks in modern bioinformatics. Perhaps even more worryingly, it is becoming harder to fill the widening gap between bioinformatics and hypothesis-driven experimental research. We acknowledge that NetR/AttR represent an overly simplistic approach to deal with heterogeneous datasets. However, and as indicated above, we developed NetR and AttR as heuristic tools for the generation of testable hypotheses based on largely unexploited datasets. In other words, NetR and AttR should not be used to answer biological questions, but to ask new ones; and we now wish to share these tools with experimental biologists at large, with the expectation that our programs will facilitate significant advances in their research efforts.

## Figures and Tables

**Figure 1 genes-10-00423-f001:**
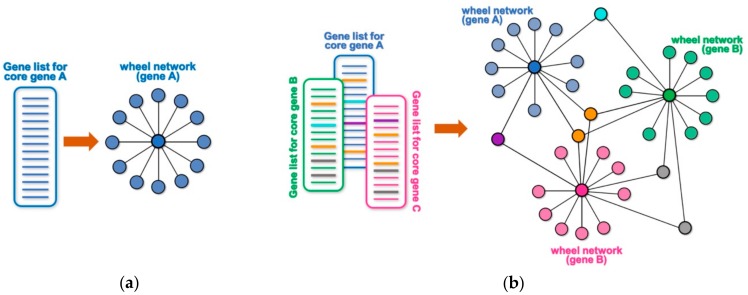
Schematic representation of wheel and NetR-integrated networks. (**a**) Any list of targets or functional partners associated to a gene or protein of interest can be schematically represented as a simple “wheel network”, where the core gene/protein is at the center of the wheel and connected to all of its targets/partners on the rim. (**b**) NetR automatically integrates several such lists of targets/partners, giving rise to wheel networks that will be interconnected by any genes/proteins that are present in more than one of the original lists, giving rise to more complex modules of hypothetical interactions among nodes in the network.

**Figure 2 genes-10-00423-f002:**
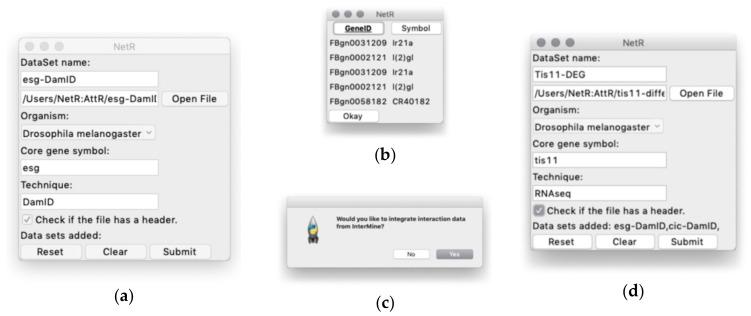
The NetR graphical user interface. (**a**) The main user interface, through which NetR users name and route the dataset to be uploaded, indicate the biological species from which it originated, and name the core gene and the technique corresponding to the dataset. (**b**) Example of the file preview window that NetR provides for users to determine which column contains the list of targets/partners of the core gene. The bold-underlined column will be used to get up-to-date identifiers for each of the genes/proteins in the uploaded dataset from InterMine, while the non-bolded, non-underlined column will be ignored. (**c**) After submitting the first dataset in a session, NetR asks users whether they want to integrate InterMine data into their network or only combine user-provided datasets. This decision is made once and applies to all datasets uploaded in a session. (**d**) Example of what the graphic interface looks like after uploading the Esg-DamID and Cic-DamID datasets (notice their listing at the bottom of the window), and just before finalizing the uploading of the Tis11-DEG one. DEG: Differentially expressed gene.

**Figure 3 genes-10-00423-f003:**
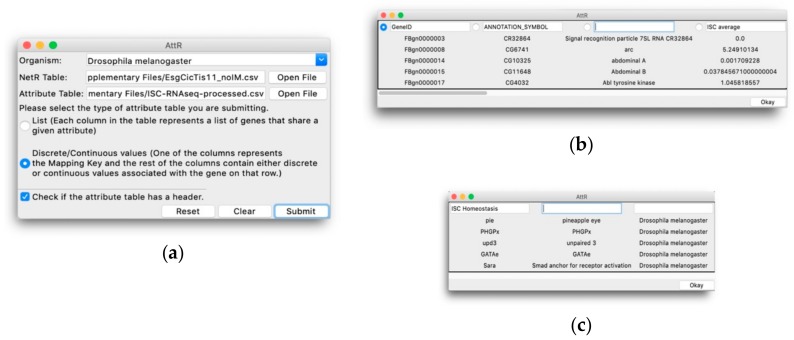
The AttR graphical user interface. (**a**) The main user interface, through which AttR users indicate the biological species that corresponds to the NetR network to be used, route the NetR and attributes files to be processed by AttR and indicate whether the attributes list corresponds to a List (**b**) or a Discrete/Continuous (**c**) type. (**b**) Preview window for a Discrete/Continuous values table, in which users identify the mapping column containing a list of genes/proteins, as well as name any other columns containing information about the genes/proteins in the mapping column that they want AttR to process. (**c**) Preview window for a List attributes table, in which the user indicates which column contains the genes/proteins associated with the corresponding attribute, which the user needs to name by typing in the corresponding window.

**Figure 4 genes-10-00423-f004:**
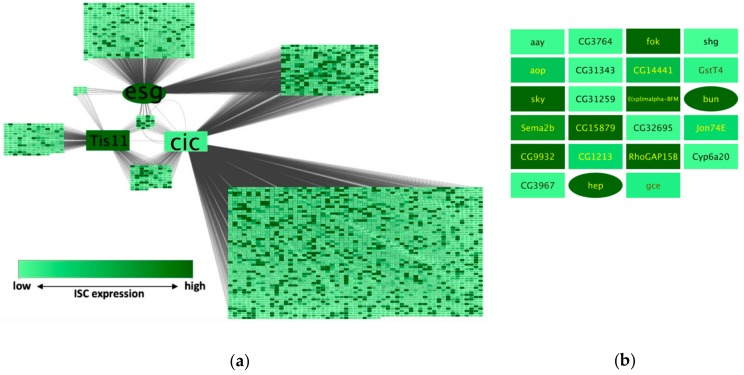
Example of a NetR / AttR network (without InterMine data integration). (**a**) A NetR/AttR network combining genome-wide data about putative transcriptional targets of the transcription factors Esg and Cic, and genes differentially expressed following the genetic manipulation of the RNA-binding protein Tis11. Darker colors represent higher expression in intestinal stem cells according to an independent study [10]. In addition, genes that are curated under the gene ontology (GO) Biological Process “Intestinal stem cell homeostasis” (for *D. melanogaster*) are represented as oval-shaped nodes (harder to notice at this scale but more easily noticeable in **b**). The core genes in this NetR/AttR network (*esg*, *cic* and *Tis11*) have been individually enlarged for presentation purposes. (**b**) The subset of genes putatively targeted by Esg, Cic and Tis11, colored and shaped to represent their intestinal stem cell (ISC) expression level and GO classification as “Intestinal stem cell homeostasis” as in (**a**).

**Figure 5 genes-10-00423-f005:**
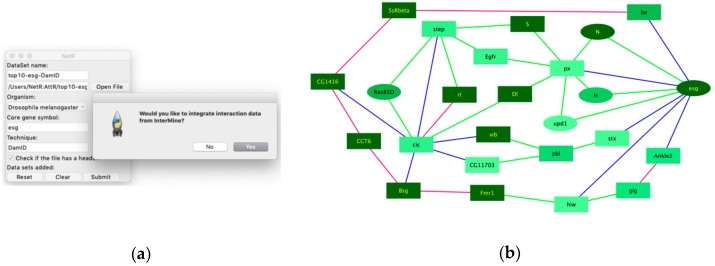
Example of a NetR/AttR network that integrates InterMine data. (**a**) To obtain additional interaction data from InterMine we answer “Yes” to the corresponding dialog box that the NetR user interface generates after uploading the first dataset in a session. (**b**) A 25-node core network generated by NetR by combining the top 10 targets of Esg and Cic while integrating additional interaction data for all the nodes as deposited in InterMine. The core network shown results from using Cytoscape functions to remove all the terminal nodes and linear paths from the original NetR network (Appendix A), as explained in Methods. Blue edges represent the user-provided DamID interactions, green and pink edges represent genetic and physical interactions, respectively, and were obtained from InterMine. The color and shape of the nodes correspond to their expression level in ISCs and their GO categorization as ISC homeostasis genes, respectively (as in Figure 4).

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
