# Peer review of "NetR and AttR, Two New Bioinformatic Tools to Integrate Diverse Datasets into Cytoscape Network and Attribute Files"

_genes, 2019, doi:10.3390/genes10060423_

Reviewer 1 Report

Halajyan and Loza-Coll present two interconnected bioinformatic tools NetR and AttR which can assist biologists to preprocess the network data and integrate them into Cytoscape recognizable input files for further visualization and downstream analysis. In this manuscript, the authors carefully demonstrated the usage of NetR/AttR with a detailed step-by-step instruction covering the whole process from installing the software to analyzing the output data. To demonstrate the utility, they used NetR/AttR to analyze a combined dataset of the genetic regulation of intestinal stem cells in Drosophila. In addition, the authors also reviewed other cutting edge tools with similar features and discussed about the uniqueness of NetR/AttR in the discussion section. Overall, the tools are very useful and I believe it will benefit the whole research community. Here, I only have some minor comments.

1) The authors attached the two Python scripts in the supplementary files which might not be a good way for updating and disseminating the software. To maximize their utility, one possible solution is to create a GitHub repository and include the URL in this manuscript so that users can obtain any updated version of both software and corresponding documentation.

2) It is very difficult to see the node labels clearly in Figure 4b and 5b. Please adjust the font color accordingly.

Author Response

Dear Reviewer 1,

Thank you very much for your feedback and suggestions. We are certainly glad to see that you found our programs to be potentially very useful to the research community, since such had always been the original intention of our efforts.

What follows is our point-by-point response to your comments and suggestions:

1) The authors attached the two Python scripts in the supplementary files which might not be a good way for updating and disseminating the software. To maximize their utility, one possible solution is to create a GitHub repository and include the URL in this manuscript so that users can obtain any updated version of both software and corresponding documentation.

We could not agree more with your suggestion. Sharing our code as open source had always been our original intention - we were simply holding back on doing so through the original submission cycle, should any major modifications have been suggested during review. We have now created a GitHub deposit and share the corresponding URL in the revised version of our manuscript (L121, https://github.com/armenhalajyan/NetRAttR). Therein, not only did we share the original code and made reasonable efforts to provide ample documentation and guidelines for users in the ReadMe section, but also included new executable files for Windows users that they can double-click to start both programs.

2) It is very difficult to see the node labels clearly in Figure 4b and 5b. Please adjust the font color accordingly.

Thank you for pointing this out. We have now changed the Cytsocape parameters for node labeling to make sure that in both figures there is a better contrast between node labels and their background color (Figure 4b and 5b; also see L230-37 and L254 of the manuscript for an updated set of instructions in Methods).

We would like to thank you once again for your appreciation of our work and your constructive suggestions.

MLC

Reviewer 2 Report

This paper presents two new software tools  NetR and AttR that are announced as a way to reuse already published data. These tools take some already preprocessed data and process it so it can be loaded to Cytoscape. The output of these tools are just some files to be used in Cytoscape.

In the discussion section, it compares these tools to other available tools. But the main difference between those tools and the paper tools is that they are all integrated in Cytoscape. That is, they can be used in Cytoscape and the output of those tools is already available in the Cytoscape environment. However, NetR and AttR are just standalone tools that output files. Subsection 2.6 dedicates a lot of lines to enumerate several steps that need to be done in order to use those output files.

I think the main weakness in this paper is that it does not present a tool that is integrated into Cytoscape and which could take care of all the loading and visualization steps. Currently, there are also other ways to do this integration besides creating a Cytoscape app. For instance, Cytoscape provides a REST API which allows to interact with Cytsocape from a Python or R script.

There is also a need to apply several preprocess steps before using NetR or AttR (subsection 2.2). But if the target users are experimental biologist with no programing background, the presented tools should also try to better help those users on those preprocessing steps. Since, for instance, calculating the average signal-to-backgound ratio might not be straight forward for experimental biologists using spreadsheet software(e.g. excel).

From my point of view, without this Cytoscape integration and other improvements, the paper and the introduced tools do not bring enough added value.

Author Response

Dear Reviewer #2,

Thank you very much for your candid feedback. Below is our point-by-point response to your comments (some of them have been re-ordered for flow):

This paper presents two new software tools NetR and AttR that are announced as a way to reuse already published data. These tools take some already preprocessed data and process it so it can be loaded to Cytoscape. The output of these tools are just some files to be used in Cytoscape.

In the discussion section, it compares these tools to other available tools. But the main difference between those tools and the paper tools is that they are all integrated in Cytoscape. That is, they can be used in Cytoscape and the output of those tools is already available in the Cytoscape environment. However, NetR and AttR are just standalone tools that output files. [...] I think the main weakness in this paper is that it does not present a tool that is integrated into Cytoscape and which could take care of all the loading and visualization steps.

We very much appreciate your recommendation that NetR and AttR should be integrated into the Cytsocape environment but would also like to share some arguments in favor of keeping our programs as standalone tools.

We believe that Cytoscape integration would limit the flexibility of the programs, both of which were conceived and produced to be open-source and easily customizable to serve additional purposes. On a very basic level, NetR takes a list of genes/proteins with often outdated identifiers and provides an up-to-date list of interactions which can be analyzed in a number of ways with or without Cytoscape. While the NetR and AttR outputs work very well with Cytoscape by design, the data in them could be useful outside of Cytoscape, particularly for more advanced users that may want to integrate them into their own bioinformatics pipelines. For example, some users may choose to only analyze the output files statistically without actually visualizing the network. They could also have other creative uses for the NetR output that we have not thought of as of yet.  Lastly, there already exist alternatives to Cytoscape (Gephi, NodeXL, GraphViz and NetworkX to name a few), and while we will continue to be strong advocates of Cytoscape, we would not want to limit users who may want to choose a different network mapping software.

Currently, there are also other ways to do this integration besides creating a Cytoscape app. For instance, Cytoscape provides a REST API which allows to interact with Cytsocape from a Python or R script.

Thank you very much for this suggestion and bringing this option to our attention. From the Cytoscape RESTful API documentation, however, it seems like the REST API would do the opposite, i.e. call Cytoscape functions into Python or R code. In other words, one could call on the REST API to bring Cytoscape features into NetR/AttR (filtering, styling changes, etc). This would allow the create of output files that already have all the desired downstream modifications to the network, or to utilize Cystoscape network analysis tools for examining the networks created by integrating datasets through NetR. Of course, we see these as very powerful options that more experienced users would like to explore; although, once again, it would require a level of coding expertise that most of the intended NetR/AttR users probably don't have. Truthfully, we were not aware of the REST API option, and we have now included some language in our revised manuscript to bring this possibility to the attention of more experienced users (L567-572). Thank you!

Subsection 2.6 dedicates a lot of lines to enumerate several steps that need to be done in order to use those output files.

Yes, we agree, subsections 2.6 and 2.7 are fairly detailed accounts of the steps to follow to use the NetR and AttR network files in Cytoscape. We have struggled with this decision ourselves: on the one hand, we felt compelled to simply direct the readers of our manuscript to import the output network and attribute files in Cytoscape. On the other hand, we considered that while many experimental biologists who might benefit from using NetR and AttR will likely know of Cytsocape, they may have never used it. And Cytsocape, for all of its potential and powerful performance, does present a learning curve. That is why we felt that we needed to offer readers, many of whom may be first-time Cytoscape users, a few initial steps to have them initiated in mapping and decorating Cytoscape networks.

There is also a need to apply several preprocess steps before using NetR or AttR (subsection 2.2). But if the target users are experimental biologist with no programing background, the presented tools should also try to better help those users on those preprocessing steps. Since, for instance, calculating the average signal-to-backgound ratio might not be straight forward for experimental biologists using spreadsheet software(e.g. excel).

It is difficult to draw a hard line between levels of programming experience among experimental biologists. Some will have formal or self-taught training in computer programming and may write their own code. Others may not code but will not be discouraged to run scripts via command line or write ad-hoc macros (e.g. in ImageJ). Some may use a variety of non-traditional programs designed specifically for a sub-discipline (phylogeny building, molecular biology workbenches and automated image analysis tools), many of which may offer built-in environments for creating a customizable pipeline. Lastly, there are those who will may not feel confident to experiment outside of traditional computer programs (e.g. MS Office, the Adobe suite and regular experimental biology websites for information retrieval). However, we do think that it is fair to assume that even the latter group of experimental biologists can perform the steps outlined in subsection 2.2 for file processing, because they all involve standard procedures involving basic Word/Excel. Most of the details in subsection 2.2 were provided mostly for reproducibility purposes. We conceived NetR and AttR to bridge that baseline level of comfort that most experimental biologists have with programs like Word and Excel with the slightly more demanding coding skills needed for updating identifiers, mapping connections between separate lists and re-formatting unstructured published data. In fact, it is this widespread comfort with a wide range of operations in programs like Word and Excel that often leads many of our colleagues to generate unstructured tables and spreadsheets in .doc, .xls or .pdf formats, which are then submitted as supplementary files with their publications and create the very problem the NetR and AttR are expected to alleviate. We should note that we had already included a paragraph explaining our stance on this point in the original version of the manuscript (L461-466 of the original manuscript, L475-480 in the revised version).

From my point of view, without this Cytoscape integration and other improvements, the paper and the introduced tools do not bring enough added value.

We intended to offer experimental biologists a tool that would allow them to exploit the wealth of underutilized published data that escape standard bioinformatic querying and retrieval, by leveraging the power of Cytoscape. In this regard, we agree that it would be sensible to integrate our programs into Cytoscape directly. However, for some of the reasons outlined above, we still think that is better to offer NetR and AttR as open source code through GitHub and let the community of more experienced users develop them into a wider array of better programs and components of their own bioinformatic pipelines. In fact, such appears to have been the evolution of many current Cytoscape apps - they began as standalone programs/websites and, as the community improved them through iterations of use and feedback, they were later incorporated as default Cytoscape Plugins/Apps.

However, we understand and very much respect your viewpoint, and we have therefore included an explicit invitation to the Cytoscape community to experiment with our programs and add any useful features, including Cytoscape integration or using the REST API as part of our code (L567-L572).

Thank you very much once again for your time and your thoughtful and candid suggestions. We hope that you may find our response and modifications to the manuscript satisfactory.

MLC  

Round  2

Reviewer 2 Report

Thank you for your detailed explanation.